# Variation in Antibiotic Prescription in High-Risk Febrile Neutropenia in Portuguese Hospitals

**DOI:** 10.3390/antibiotics13090822

**Published:** 2024-08-30

**Authors:** Marta Freitas, Paulo Andrade, Ricardo Pinto, Fernanda Trigo, Ana Azevedo, Francisco Almeida

**Affiliations:** 1Faculdade de Medicina da Universidade do Porto, Alameda Professor Hernâni Monteiro, 4200-039 Porto, Portugal; up201807440@g.uporto.pt (M.F.); a.oliveira@ulssjoao.min-saude.pt (A.A.); 2Unidade de Prevenção e Controlo de Infeção e Resistências aos Antimicrobianos, Centro de Epidemiologia Hospitalar, Unidade Local de Saúde São João, 4200-319 Porto, Portugal; paulo.andrade@ulssjoao.min-saude.pt; 3Serviço de Doenças Infecciosas, Unidade Local de Saúde São João, 4200-319 Porto, Portugal; 4Serviço de Hematologia Clínica, Unidade Local de Saúde São João, 4200-319 Porto, Portugal; rj.pinto@ulssjoao.min-saude.pt (R.P.); ftrigo@ulssjoao.min-saude.pt (F.T.); 5Centro de Epidemiologia Hospitalar, Unidade Local de Saúde São João, 4200-319 Porto, Portugal; 6Laboratório Para a Investigação Integrativa e Translacional em Saúde Populacional (ITR), EPIUnit, Instituto de Saúde Pública da Universidade do Porto, Universidade do Porto, 4050-600 Porto, Portugal

**Keywords:** febrile neutropenia, high risk, antimicrobial stewardship, survey, guidelines

## Abstract

Introduction: Febrile neutropenia (FN) is a potentially severe entity, particularly in hemato-oncologic patients who have higher incidence of colonization with multidrug-resistant bacteria. Discrepancies among guidelines contribute to divergence in antimicrobial practices. Our objective was to assess the variation of practices in antimicrobial therapy in high-risk FN among Portuguese hematologists. Methods: We conducted a cross-sectional study through the implementation of an online survey, open to all clinical hematologists in the country. To characterize practice patterns regarding critical elements in FN management, three clinical vignettes were designed to describe typical situations where narrow-spectrum empiric antibiotics (vignette 1), short-course therapy (vignette 2) and de-escalation (vignette 3) could be performed. The remaining questions characterized clinical experience, department size, and differentiation and decision-making process regarding FN antibiotic therapy. Results: The survey yielded 31 responses from 11 hospitals across four regions. All respondents opted for empiric narrow-spectrum antibiotics, 22.6% opted for short-course therapy (mostly senior specialists from larger settings) and 35.5% for de-escalation (mostly young specialists). Availability of an FN protocol seemed to favor both approaches. These findings should be complemented by qualitative assessments of barriers to best practices and should support the need for interventions to improve antibiotic use in febrile neutropenia.

## 1. Introduction

Febrile neutropenia (FN) is a frequent clinical entity in oncological patients, with most cases occurring in the context of intensive chemotherapy [1,2,3]. FN occurs in 10–50% of patients undergoing intensive chemotherapy for solid tumors and >80% in those with hematologic malignancies [3,4,5]. 

Episodes of febrile neutropenia with expected duration of neutropenia of at least 8 days, with an absolute neutrophil count (ANC) < 100 cells/μL or with Multinational Association of Supportive Care in Cancer (MASCC) score < 21 are classified as high risk [6,7,8,9,10]. Mortality in high-risk febrile neutropenia ranges from 7.4 to 11% in non-complicated high-risk FN to up to 50% if there is sepsis or septic shock at presentation [7,11,12,13]. Additionally, the treatment of acute leukemia and procedures such as autologous and allogeneic stem cell transplantations (SCT) pose increased risk of FN and potentially fatal complications, with FN mortality rates ranging from 10% in autologous SCTs, 20–26% within the first two months of acute myeloid leukemia (AML), and up to 80% during graft failure [6].

Due to the potential severity of this syndrome, prolonged durations of empirical broad-spectrum antibiotics are frequent. However, it is recognized that prolonged and presumably unnecessary exposure to broad-spectrum antibiotics leads to an increased risk of infections caused by MDR agents such as ESBL-producing gram-negative bacteria, carbapenem-resistant Enterobacterales, VRE, MRSA, and Clostridium difficile infections as well as predisposition for fungal infections, with growing incidence of fluconazole-resistant Candida strains (e.g., Candida krusei and Candida glabrata) being reported [6,8]. This risk is particularly high in patients with hematologic malignancies, who have a higher risk of colonization by MDR organisms due to frequent healthcare contact such as day hospital visits and hospitalizations, frequent infections and antibiotic therapy, and alterations in endogenous bacterial microbiome in the context of chemotherapy-induced mucositis [14,15]. Antimicrobial stewardship programs focusing on reducing inappropriate antibiotic exposure in hospitalized patients, alongside curbing excessive community and veterinary use of antimicrobials, is a priority of many scientific societies such as the World Health Organization [16].

International guidelines such as the ones issued by the UK National Institute for Health and Care Excellence (NICE, 2012 [5]), the European Society for Medical Oncology (ESMO, 2016 [6]), the American Society of Clinical Oncology (ASCO, 2018 [7]), the 4th European Conference on Infections in Leukemia (ECIL-4, 2013 [8]), the Infectious Diseases Working Party of the German Society of Hematology and Medical Oncology (AGIHO, 2017 [9]), and the Infectious Diseases Society of America (IDSA, 2011 [10]) have clear recommendations on antibiotic use in order to achieve a balance between adequately managing these potentially severe infections and avoiding antibiotic overuse. An escalation strategy beginning with narrow-spectrum choices such as piperacillin-tazobactam is generally recommended for patients without risk factors for MDR infections, complicated presentation, clinical instability, or sepsis. Similarly, it is consensual among these recommendations that empirical antibiotic therapy should be de-escalated to target specific pathogens once they are identified, and susceptibility profiles are available. However, there are different recommendations on the optimal duration of antibiotic therapy: some guidelines suggest discontinuing antibiotics after a specified period of afebrile status, irrespective of ANC recovery, while others advocate continuation until neutrophil recovery is confirmed (Table 1) [5,6,7,8,9,10].

In Portugal, numerous medical centers manage cases of febrile neutropenia. However, there are no national guidelines for the management of this entity and no data on uniformity of antibiotic practices between different institutions. 

This study aimed to evaluate the variation in practices regarding the choice of empirical antibiotic therapy, treatment duration, and decisions regarding AST-guided de-escalation, through questionnaire-based assessments of hematologists’ treatment approaches, in different Portuguese hospitals. 

## 2. Results

In 7 weeks, we received 31 responses from hospitals belonging to four different regional health administrations: North (n = 18), Lisbon (n = 7), the Center (n = 5), and Azores (n = 1) (Figure 1).

Among respondents, about half were specialists for over five years, followed by over one third who were residents. Approximately half of the participants managed over 10 episodes of FN monthly. About 40% of our sample reported individual decision making, followed by over 35% who had department meetings for case discussions, and one quarter who consulted with a colleague for FN treatment decisions (Table 2).

Out of the eleven department directors contacted, six did not respond to the survey. As a result, the section regarding department organization aimed at these directors was left unanswered. The remaining five who responded to this section allowed inclusion of department or unit characterization data for most participants, totaling 18 out of 31 respondents (Figure 2).

Most of the respondents (61%) worked in large departments and most participants (72%) had access to a local FN protocol. Half reported working in centers who admitted patients for allogeneic transplant, 61% for autologous transplant, and 90% of departments performed induction chemotherapy in AML patients (Table 3).

In vignette 1, every participant opted for narrower-spectrum empiric options; therefore, an “escalation” approach (Figure 3a). In vignette 2, when confronted with three different antibiotic duration regimens, less than one quarter of prescribers immediately discontinued antibiotic therapy, 25% (n = 8) chose to extend antibiotic administration until ANC recovery, and half chose to extend the antibiotic therapy for at least 7 days (Figure 3b). Finally, in vignette 3, most physicians (61%) would maintain the 4th generation cephalosporin included in the empirical regimen but discontinue the combined aminoglycoside, whereas only 35% of prescribers would discontinue both antibiotics and start amoxicillin-clavulanate, according to the AST (Figure 3c).

There was a marginal difference in responses to vignette 2 according to individual factors, with senior specialists, those handling over 10 FN episodes monthly and those reviewing decisions in department-meeting reporting more often antibiotic discontinuation. Similarly, we found a minimally higher frequency of short-course antibiotic therapy among respondents from large departments. In settings where a high-risk FN protocol was available, 38.5% of respondents chose to discontinue antibiotics (n = 5) versus 20% in settings without a high-risk FN protocol (n = 1) (Table 4).

Regarding clinical vignette 3, de-escalation according to AST was more frequently reported by junior specialists (80% versus 20% de-escalation by senior specialists), by hematologists treating fewer than 2 episodes of FN monthly and by those who mostly decide individually on antibiotics for high-risk FN. Additionally, physicians working in small departments, those with access to an existing high-risk FN protocol and attendants working in hospitals with less valencies, reported AST-guided de-escalation more often (Table 5).

## 3. Discussion

Our study found variation in practices regarding antibiotic duration and de-escalation guided by ASTs, which was present in all categories of clinical experience and department size assessed by our survey. All participants opted for narrower-spectrum antibiotic therapy in a non-complicated presentation of high-risk FN, in accordance with European and IDSA guidelines [5,6,7,8,9,10]. 

Regarding antibiotic duration, when responders were asked to choose the optimal antibiotic course for a stable neutropenic AML patient, post-chemotherapy, with over 48 h of defervescence and symptom resolution, only a minority suspended treatment, which would be supported by European guidelines with the potential to reduce antibiotic use (ECIL-4 and NICE) [5,8]. Most opted for an antibiotic regimen of at least 7 days, which, in this case, would result in a higher antibiotic consumption. About a quarter chose to extend treatment until ≥ 500 cells/μL, which, despite being endorsed by American IDSA guidelines [10], would lead to longer treatment durations. Time to recovery neutrophil after chemotherapy is usually longer than a week, and in situations such as stem-cell transplant, median durations of two to four weeks are often described [17,18].

These results are in line with a previous survey by Verlinden et al. where only 17.6% of respondents from 170 European hospitals would stop antibiotic therapy before day 7, in fever of unknown origin. In fact, a third of the responses from this study described treatment continuation until ANC recovery, while 47.8% would opt for a duration of 7–14 days [19]. 

AST-guided de-escalation, evaluated in vignette 3, is recommended by most international societies. However, this approach was chosen by only one third of respondents in our survey. In the survey by Verlinden et al., de-escalation was performed by 76.8% of the participating institutions, if there was a positive blood culture with a susceptible pathogen with uncomplicated presentation [19]. 

De-escalation has shown to be a safe practice in severe infections [20,21] including high-risk NF [3]. Additionally, increasing adherence to de-escalation practices in high-risk NF is associated with a decrease of broad-spectrum antibiotic use [3,22]. 

We found different trends of responses between junior and senior doctors to the questions regarding discontinuation and de-escalation of antibiotics. A qualitative study by Nilsson and Pilhammar tried to characterize medical decision making by interviewing 9 senior and 9 junior internal medicine physicians to recall and describe self-chosen events that happened in clinical practice relating either positively or negatively to their respective professional action. They suggest that junior doctors are mostly influenced by theoretical knowledge and by the use of guidelines or other credible sources of information, while having less confidence in autonomous decision making. Conversely, senior colleagues often use theoretical knowledge as complementary to their own experience of different cases and clinical courses. De-escalation after microbiological isolation, which is consistently recommended across different guidelines, was chosen more often by residents or junior specialists when compared with senior specialists. The same pattern was not found for early discontinuation of antibiotics, with senior physicians discontinuing antibiotic treatment after 48 h of apyrexia more often when compared to their junior counterparts. It is possible that the absence of consensus in the guidelines makes early discontinuation a less linear decision in clinical practice, requiring clinical experience and capacity to argument in favor of such choice [23].

The highest percentage of respondents who chose short-course therapy was found in departments with FN protocols, which was double of what was seen in the remaining departments. Similarly, departments with available FN protocols also had a higher proportion of respondents choosing AST-based de-escalation. It is expected that locally designed protocols based on international guidelines will lead to a more uniform approach and to a higher adhesion to recommended practices. Previous studies have shown success in reducing antibiotic use in clinical hematology departments through interventions which included the implementation of local protocols in the form of visual decision algorithms after discussion and revision in multidisciplinary meetings between hematology and antimicrobial stewardship teams, supported by regular meetings, for review and feedback of antibiotic prescription in hemato-oncologic patients with infection [19].

Although this study identifies areas with potential for improvement, actual antimicrobial stewardship interventions to improve care should take contextual barriers into account and should involve local experts in implementation. One successful example is a study by Horo et al., describing a quality-improvement intervention which achieved a reduction of deviation from best practices from 71% to 27.3% in the empiric approach of FN through implementation of a clinical algorithm. The algorithm was elaborated through a modified Delphi approach which included experts within the institution. Implementation was initiated with focus group meetings to identify barriers for local optimal use of antibiotics for FN and dissemination was performed using a plan–do–study–act cycle. Input on the intervention was collected monthly leading to frequent revisions of the clinical algorithm [24]. Further studies in our setting focusing on the qualitative assessment of barriers and facilitators of adhesion to best practices in FN would be an important step towards achieving a more uniform adhesion to best practices.

In interpreting the findings of our study, several limitations should be considered. Firstly, the small sample size may limit the interpretability and generalizability of our results. For example, in departments lacking facilities for SCTs or chemotherapy in AML patients, drawing conclusions regarding higher rates of short-term antibiotic treatment and AST-guided de-escalation is challenging due to the limited representation, with only two participants from such hospitals. Also, our results may not capture the full reality of antimicrobial practices for FN in our national setting. We were not able to reach the totality of hematology departments and had a low return rate, despite achieving a participation of 10% of all hematology specialists and residents registered in the national medical association. Additionally, the presentation of only one clinical case for each scenario may not fully capture the complexity of real-world practice, although we intentionally selected cases that exemplified paradigmatic cases with little ambiguity for application of guideline recommendations. Also, physicians’ perceptions of their own practices may differ from actual practice. In particular, guideline adherence has been described to be overestimated in surveys designed to assess the quality of clinical care, at least in part due to social desirability bias [25]. 

## 4. Materials and Methods

### 4.1. Study Design and Participants

A web-based cross-sectional survey was conducted in various Portuguese hospitals, aimed at clinical hematologists. The main objective was to describe the variation in antibiotic management practices regarding choice of empirical antibiotic therapy, treatment duration and de-escalation in high-risk febrile neutropenia.

The project was presented to the heads of hematology units or departments from different Portuguese hospitals along with a link to the online survey, requesting divulgation among clinical hematologists in the unit or department. A description of the project and a link to the survey was also posted on the official website of the Portuguese Medical Association, Ordem dos Médicos. 

The questionnaire was written in Portuguese and was open for 7 weeks (between 18 January and 6 March 2024). Several reminders were sent to each department during weeks 2–7.

Participation was voluntary and anonymous. Participants were informed about the purpose, methods, and intended uses of the research on the first page. Before starting the online survey, participants confirmed their acceptance to participate. The ULSSJ ethics committee approved the study (number 430/2023).

### 4.2. Data Collection

The questionnaire consisted of two parts. The first part focused on the characterization of the participants and respective department or unit. Some questions related to each department or unit’s characteristics were available only for the head of the department or unit (points 2–4): Name of participating hospital and corresponding Regional Health Administration;Type of facility (department or unit);Size of the respective department or unit, regarding the number of beds and the number of attending physicians;Department or unit experience in high-risk FN estimated first by the quantity of FN episodes treated on the ward per month and second by their differentiation in induction chemotherapy for AML or autologous or allogeneic bone marrow transplantation;The availability of an internal protocol for antibiotic therapy in high-risk FN;Description of the usual decision-making context regarding antibiotic therapy in high-risk FN (discussion in a department or unit meeting, discussion with colleagues our individual decision-making);Individual experience of the attending physician, inferred from questions regarding the level of training (specialist or resident); the number of years as a specialist (<5 years or ≥5 years); and the number of high-risk FN treated per month under direct care responsibility.

The second part was composed of three clinical scenarios developed by a specialist in clinical hematology (RP) and a specialist in infectious diseases (PA), experienced in managing FN (Figure 4).

Clinical vignette 1 illustrated a scenario where an “escalation strategy” starting with narrower-spectrum therapy would be recommended by most guidelines. Respondents were presented with options representing both narrower-spectrum (escalation strategy) and broader-spectrum (de-escalation strategy) choices, which were categorized accordingly;In clinical vignette 2 responders could opt between an antibiotic-sparing short-course treatment, irrespective of ANC, supported by ECIL-4 and NICE guidelines, and two other regimens: a treatment duration of at least 7 days, commonly used in clinical practice, or treatment until ANC ≥ 500 cells/μL, supported by IDSA guidelines;Clinical vignette 3 depicts a situation where de-escalation would be appropriate according to most guidelines. Respondents could opt to de-escalate based on culture isolation and AST, solely modify the regimen by discontinuing broader spectrum antibiotics, or maintain the broad-spectrum empiric regimen.

We defined large departments as departments or units with over 20 beds, over 10 attending hematologists, and handling over 20 cases of febrile neutropenia monthly.

### 4.3. Data Analysis

Data were collected in Google Forms^®^ and stored in Microsoft Excel 2013^®^. Data were exported to IBM SPSS Statistics 21^®^ for analysis.

Individual features of clinical hematologists were grouped into the following categories: professional experience (residents, specialists for less than 5 years, specialists for more than 5 years); number of FN episodes assisted under direct responsibility per month (<2/month, 2–10/month, >10/month); decision-making strategy (performed at the individual discretion of the attending physician, reviewed with a clinical hematology colleague, reviewed during a department meeting). 

Features of the hematology department were grouped as follows: department size (large hematology department, small hematology department); the differentiation in induction chemotherapy for AML and in autologous or allogeneic bone marrow transplantation (AML induction ChT and BM transplant, AML induction ChT but no BM Transplant, no AML induction ChT and no BM Transplant); the existence of high-risk FN (without an established high-risk FN protocol, with an established high-risk FN protocol).

In clinical vignette 1, options 1–3 were categorized as “narrower-spectrum options” and the remaining as “broader-spectrum options”. In clinical vignette 2, option 1 was categorized as “short course treatment” and the remaining as “long course treatment”. In clinical vignette 3, option 3 was categorized as “AST-guided de-escalation” and options 1–2 as “non de-escalation despite AST”.

We described the frequency of the different responses in clinical vignettes first in the whole sample and afterwards in subgroups according to the categories of individual features of clinical hematologists or to features of the hematology department.

## 5. Conclusions

We observed diversity in FN management practices across Portuguese hospitals. We also found that a significant proportion of practices do not align with guidelines and may result in higher antibiotic consumption. 

Our study identifies some areas with room for improvement regarding antimicrobial stewardship objectives such as reducing duration and spectrum of antibiotic therapy [26]. Although barriers and facilitators to optimal clinical practice should be locally assessed before designing interventions, our results show that availability of local guidelines are features of the setting that better align with recommended practices. Patient safety and the need to adapt decisions to the individual patient should nonetheless be safeguarded.

## Figures and Tables

**Figure 1 antibiotics-13-00822-f001:**
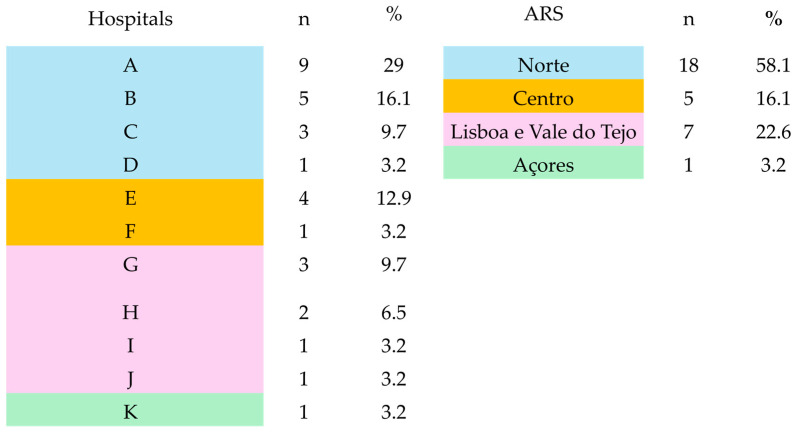
Regional distribution of survey responses. ARS—Regional Health Administration areas.

**Figure 2 antibiotics-13-00822-f002:**
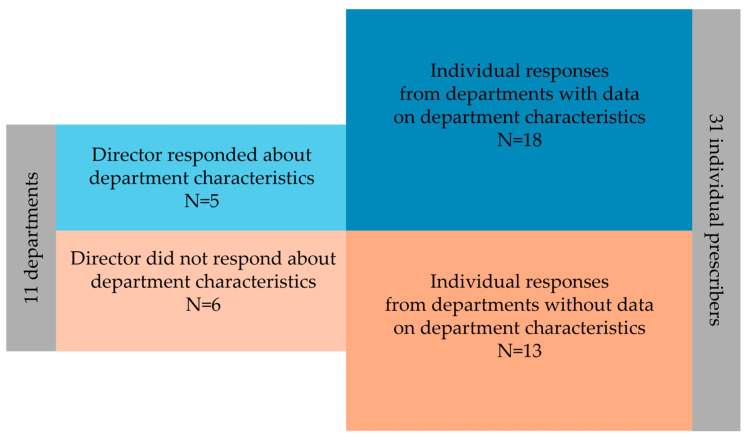
Included departments with and without data on department characteristics depending on whether each head of department was able to respond to the survey.

**Figure 3 antibiotics-13-00822-f003:**
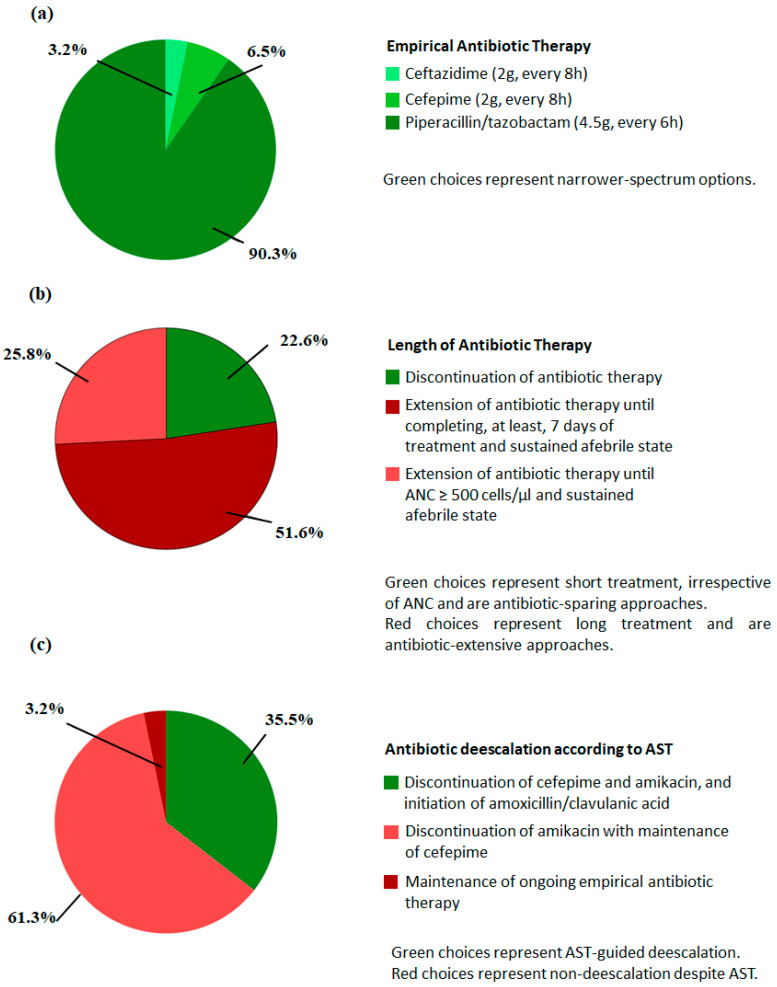
Survey responses to the three clinical vignettes regarding (**a**) choice of empirical therapy, (**b**) duration of antibiotic treatment, and (**c**) AST-guided de-escalation.

**Figure 4 antibiotics-13-00822-f004:**
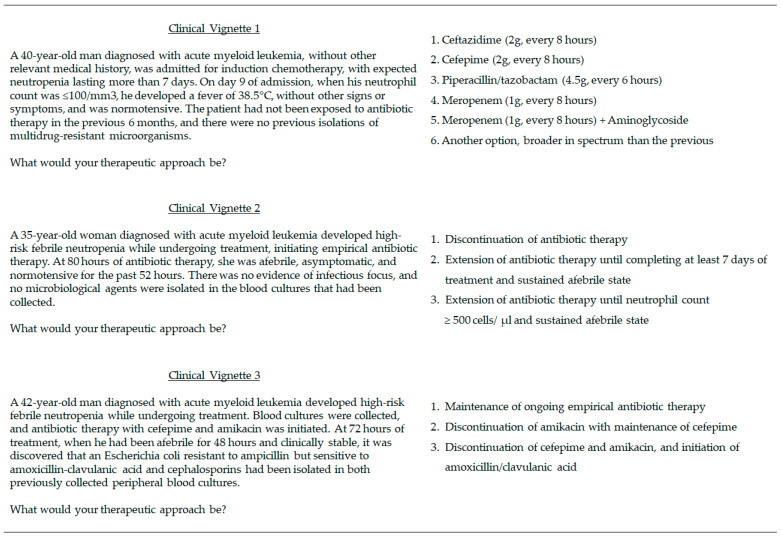
Clinical vignettes depicting typical decisions regarding antibiotic use in high-risk FN: (1) choice of empirical therapy; (2) duration of antibiotic treatment and (3) AST-guided de-escalation.

**Table 1 antibiotics-13-00822-t001:** Guideline recommendations on antibiotic therapy for high-risk FN.

Guideline	Empirical Antibiotic Therapy for Patients without Risk Factors for MDR Infections, Complicated Presentation, Clinical Instability, or Sepsis	Discontinuation of Antibiotic Therapy Dependent on Absolute Neutrophil Count (ANC)?	Minimal Duration of Antimicrobial Therapy	Microbiologically Directed De-Escalation
NICE	Escalation approach	No	Discontinuation when neutropenic sepsis has responded to treatment.	Yes
ECIL-4	Escalation approach	No	3 days if the patient has been afebrile and clinically stable for ≥48 h.	Yes
ESMO	Escalation approach	No	5–7 days if afebrile and symptom free. Except in high-risk cases where antibiotics are continued for 10 days or until ANC recovery.	Yes
AGIHO	Escalation approach	No	7 days after resolution of fever	Yes
IDSA	Escalation approach	Yes	Dependent on ANC recovery (≥500 cells/μL)	Yes

**Table 2 antibiotics-13-00822-t002:** Characteristics of clinical hematologists who responded to the survey.

	n	%
Professional Experience		
Resident	11	35.5
Specialist < 5 years	5	16.1
Specialist ≥ 5 years	15	48.4
FN assisted */month		
0–1	5	16.1
2–5	8	25.8
6–10	4	12.9
11–20	7	22.6
>20	7	22.6
Decision Making		
Individually made by assistant doctor	12	38.7
Discussed with hematology colleague	8	25.8
Department Meeting	11	35.5

* Regarding to FN episodes assisted under direct care responsibility of the clinical hematologist.

**Table 3 antibiotics-13-00822-t003:** Characteristics of clinical hematology departments.

	n	%
No. Beds		
variable, allocated to other departments	7	38.9
>20	11	61.1
No. Physicians		
<5	7	38.9
>10	11	61.1
AML Inpatient Care		
no	2	11.1
yes	16	88.9
Autologous BM Transplant		
no	7	38.9
yes	11	61.1
Allogenic BM Transplant		
no	9	50
yes	9	50

**Table 4 antibiotics-13-00822-t004:** Responses to clinical scenario 2 (regarding length of antibiotic therapy) according to individual characteristics (left) and department features (right).

	Characteristics of Clinical Hematologists				Characteristics of Hematology Departments		
Length of Antibiotic Therapy			Short treatment Irrespective of ANC	Long treatment: 7-day length or Until ANC ≥ 500 cells/μL		Short treatment Irrespective of ANC	Long treatment: 7-day length or Until ANC ≥ 500 cells/μL
	Professional Experience				Department Size		
	Resident		18.2% (2)	81.8% (9)	Large	36.4% (4)	63.6% (7)
	Specialist < 5 years		20% (1)	80% (4)	Small	28.6% (2)	71.4% (5)
	Specialist ≥ 5 years		26.7% (4)	73.3% (11)			
	FN Assisted/Month				High-Risk FN Protocol		
	<2		20% (1)	80% (4)	No	20% (1)	80% (4)
	2–10		16.7% (2)	83.3% (10)	Yes	38.5% (5)	61.5% (8)
	>10		28.6% (4)	71.4% (10)			
	Decision Making				Hospitals Performing ChT for AML/ Hospitals Performing BMT		
	Individually made by assistant doctor		16.7% (2)	83.3% (10)	Yes/Yes	36.4% (4)	63.6% (7)
	Discussed with hematology colleague		25% (2)	75% (6)	Yes/No	20% (1)	80% (4)
	Department Meeting		27.3% (3)	72.7% (8)	No/No	50% (1)	50% (1)
		Total	22.6% (7)	77.4% (24)		33.3% (6)	66.7% (12)

**Table 5 antibiotics-13-00822-t005:** Responses to clinical scenario 3 (regarding AST-guided de-escalation) according to individual characteristics (left) and department features (right).

	Characteristics of Clinical Hematologists				Characteristics of Hematology Departments		
De-escalation According to AST			AST-guided de-escalation	Non de-escalation despite AST		AST-guided de-escalation	Non de-escalation despite AST
	Professional Experience				Department Size		
	Resident		36.4% (4)	63.6% (7)	Large	36.4% (4)	63.6% (7)
	Specialist < 5 years		80% (4)	20% (1)	Small	57.1% (4)	42.9% (3)
	Specialist ≥ 5 years		20% (3)	80% (12)			
	FN Assisted/Month				High-Risk FN Protocol		
	<2		60% (3)	40% (2)	No	40% (2)	60% (3)
	2–10		33.3% (4)	66.7% (8)	Yes	46.2% (6)	53.8% (7)
	>10		28.6% (4)	71.4% (10)			
	Decision Making				Hospitals Performing ChT for AML/ Hospitals Performing BMT		
	Individually made by assistant doctor		41.7% (5)	58.3% (7)	Yes/Yes	36.4% (4)	63.6% (7)
	Discussed with hematology colleague		37.5% (3)	62.5% (5)	Yes/No	40% (2)	60% (3)
	Department meeting		27.3% (3)	72.7% (8)	No/No	100% (2)	0
		Total	35.5% (11)	64.5% (20)		44.4% (8)	55.6% (10)

## Data Availability

The data presented in this study are available on request from the corresponding author due to ethical reasons.

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
