# Peer review of "Variation in Antibiotic Prescription in High-Risk Febrile Neutropenia in Portuguese Hospitals"

_antibiotics, 2024, doi:10.3390/antibiotics13090822_

Round 1

Reviewer 1 Report

Comments and Suggestions for Authors

Authors have conducted a cross-sectional study using an online survey; open to all clinical hematologists in Portugal. To characterize practice patterns in FN management, they have designed three clinical vignettes depicting typical situations involving narrow-spectrum empiric antibiotics (vignette 1), short-course therapy (vignette 2), and de-escalation (vignette 3). Although the survey study has not wet lab experiments but it bears critical information to the relevant society. I suggest authors to revise the title as “Variation in Antibiotic Prescribing Practices for High-Risk Febrile Neutropenia Among Portuguese Hematologists"”

Please clearly explain in the introduction that >  are there any specific gaps in the current guidelines that  lead to divergences in antimicrobial practices?

Results: What are the demographics of the respondents (e.g., age, years of experience, hospital size)?

Discussion: Please explain on how does the availability of an FN protocol influence the choice of therapy?

Conclusion: please align the conclusion with the points that qualitative assessments are suggested to understand barriers to best practices.. and added that what types of interventions could improve antibiotic use in febrile neutropenia based on these findings?

the authors need to correct the Table format and font size. The Table seems a very early draft.

they survey study may have an impact on the relevant society. Therefore it can be accepted for publication

Comments on the Quality of English Language

NA

Author Response

We appreciate the reviewer's constructive criticism and the positive comments and we hope our work does indeed have a positive impact in society.

I suggest authors to revise the title as “Variation in Antibiotic Prescribing Practices for High-Risk Febrile Neutropenia Among Portuguese Hematologists"”-

We agree with the suggestion and have changed the title in the revised manuscript

Please clearly explain in the introduction that >  are there any specific gaps in the current guidelines that  lead to divergences in antimicrobial practices?

Thank you for your input. We believe that, regarding the domains of antbiotic prescription we tried to evaluate in our article, the main issues regarding the guidelines are the absence of consensus regarding duration of treatment. We have expanded our discussion on that issue in the introduction.

Results: What are the demographics of the respondents (e.g., age, years of experience, hospital size)?

We agree that a demographic characterization of our sample is relevant. We opted not to collect the specific age or gender of respondents to preserve anonymity, especially in small departments. The data we have on years of experience and size of hospital are presented in tables 2 and 3

Discussion: Please explain on how does the availability of an FN protocol influence the choice of therapy?

We believe that protocols designed locally and based on international guidelines will lead to a more uniform approach and to a higher adhesion to recommended practices. We have added a line trying to clarify this idea in the text.

Conclusion: please align the conclusion with the points that qualitative assessments are suggested to understand barriers to best practice and added that what types of interventions could improve antibiotic use in febrile neutropenia based on these findings?

Thank you for your insight. We agree that our initial conclusions could be improved We have expanded the conclusion with your recommendations and we believe that it improves the message.

The authors need to correct the Table format and font size. The Table seems a very early draft.

Thank you. We have tried to improve the tables’ design and we hope these are easier to read

Reviewer 2 Report

Comments and Suggestions for Authors Interesting survey based on the ever-minimum percentage of collaborative responses. In the introduction, more extensive reference could be made to the activities to contain antibiotic pollution (also mentioning the high veterinary quota) by WHO and Scientific Societies of other Specialties [i.e. FEOph Symposium SOI 2023, Rome: “Anaesthesia, Antibiotics, Anti-Inflammatory in Ophthalmic Surgery”], etc.  Well presented work

Author Response

In the introduction, more extensive reference could be made to the activities to contain antibiotic pollution (also mentioning the high veterinary quota) by WHO and Scientific Societies of other Specialties [i.e. FEOph Symposium SOI 2023, Rome: “Anaesthesia, Antibiotics, Anti-Inflammatory in Ophthalmic Surgery”], etc.  Well presented work

We have expanded the introduction to include the reviewer's suggestion.

We appreciate the positive comments and are glad that the reviewer enjoyed our article. 

Reviewer 3 Report

Comments and Suggestions for Authors

The authors reported very interesting data obtained from revisiting attitude in Febrile Neutropenia patients in a number of Hospitals in Portugal.

Despite everything could be improved, in this case, data compilation is of high medical value as FN use to represent one situation in front of which, taking therapeutical decisions is always difficult as not much information exist so far.

Methodologically correct, thoughtful discussion and results´s interpretation  drives to interesting and relevan conclusions that might help the management of patients with FN in the immediate future.

Author Response

We thank the reviewer for the positive feedback and we are glad to see your work being appreciated,